# Exploring the Fate of Antibody-Encoding pDNA after Intramuscular Electroporation in Mice

**DOI:** 10.3390/pharmaceutics15041160

**Published:** 2023-04-06

**Authors:** Marie-Lynn Cuypers, Nick Geukens, Kevin Hollevoet, Paul Declerck, Maarten Dewilde

**Affiliations:** 1Laboratory for Therapeutic and Diagnostic Antibodies, Department of Pharmaceutical and Pharmacological Sciences, KU Leuven-University of Leuven, O&N II Herestraat 49 Box 820, 3000 Leuven, Belgium; 2PharmAbs-The KU Leuven Antibody Center, KU Leuven-University of Leuven, O&N II Herestraat 49 Box 820, 3000 Leuven, Belgium

**Keywords:** antibody gene transfer, plasmid DNA, intramuscular electroporation, DNA localization, DNA/RNA quantification, DNA scope

## Abstract

DNA-based antibody therapy seeks to administer the encoding nucleotide sequence rather than the antibody protein. To further improve the in vivo monoclonal antibody (mAb) expression, a better understanding of what happens after the administration of the encoding plasmid DNA (pDNA) is required. This study reports the quantitative evaluation and localization of the administered pDNA over time and its association with corresponding mRNA levels and systemic protein concentrations. pDNA encoding the murine anti-HER2 4D5 mAb was administered to BALB/c mice via intramuscular injection followed by electroporation. Muscle biopsies and blood samples were taken at different time points (up to 3 months). In muscle, pDNA levels decreased 90% between 24 h and one week post treatment (*p* < 0.0001). In contrast, mRNA levels remained stable over time. The 4D5 antibody plasma concentrations reached peak levels at week two followed by a slow decrease (50% after 12 weeks, *p* < 0.0001). Evaluation of pDNA localization revealed that extranuclear pDNA was cleared fast, whereas the nuclear fraction remained relatively stable. This is in line with the observed mRNA and protein levels over time and indicates that only a minor fraction of the administered pDNA is ultimately responsible for the observed systemic mAb levels. In conclusion, this study demonstrates that durable expression is dependent on the nuclear uptake of the pDNA. Therefore, efforts to increase the protein levels upon pDNA-based gene therapy should focus on strategies to increase both cellular entry and migration of the pDNA into the nucleus. The currently applied methodology can be used to guide the design and evaluation of novel plasmid-based vectors or alternative delivery methods in order to achieve a robust and prolonged protein expression.

## 1. Introduction

Monoclonal antibody (mAb)-based therapies have revolutionized the treatment paradigms in a multitude of disease areas [1,2]. With antibody gene transfer, the nucleotide sequence that encodes the therapeutic protein is administered to patients, instead of the therapeutic protein itself. The human body is turned into a mAb “factory”, resulting in the production and secretion of mAbs into the bloodstream for a prolonged period of time [3]. This treatment strategy has multiple advantages over conventional antibody therapy. The less frequent administrations and gradual in vivo mAb production and buildup are anticipated to improve patient comfort and safety. Compared to conventional recombinant proteins, nucleic-acid based therapies can be rapidly designed and produced at large scale, saving both time and resources. The stability of pDNA at room temperature negates the current need for cold-chain storage and shipment, facilitating dissemination. Overall, these advantages can increase the accessibility and implementation of mAb therapy. 

Our research group previously demonstrated proof of concept for DNA-based intramuscular gene transfer in mice, hamsters and sheep [4,5,6,7,8]. pDNA is a favorable expression platform in terms of immunogenicity, biosafety, and payload capacity compared to viral vectors. However, it has a low transfection efficiency, which generally results in lower protein expression levels [3,9]. Physical methods, such as electroporation, are therefore required to overcome the low transfection efficiency [3]. Hence, intramuscular pDNA injection is followed by the application of electrical pulses, leading to temporal cell membrane poration, allowing uptake of the pDNA in the muscle cells [10]. Expression levels improve after this procedure, but are still lower compared to viral vector delivery. Despite its lower protein expression, DNA-based antibody gene transfer has shown preclinical efficacy and is currently under clinical evaluation (NCT01138410 and NCT03831503).

The current pharmacokinetic (PK) profile of mAbs expressed in vivo in mice after DNA-based delivery is typically characterized by a steady increase in plasma mAb concentration in the first two to three weeks. Peak levels reach single-digit to double-digit µg/mL in mice. However, thereafter a decline in mAb levels is typically observed, although mAb remains detectable for many months after the initial pDNA injection [4]. This is comparable to the levels observed in sheep, where peak plasma levels reached 5 µg/mL after administration of 12 mg pOVAC in the absence of anti-drug antibodies. Sheep are more similar to humans in terms of body weight, musculature, and blood volume, and therefore these data provide valuable insights towards clinical translation [7]. For a broad implementation of DNA-based mAb delivery, a more stable expression profile at high mAb concentrations would be beneficial. Ideally, these levels need to be maintained above or around 5–10 μg/mL for a longer time, in line with trough levels for most therapeutic mAbs in humans [11].

The limited mAb expression that can be attained in vivo—only peak levels reaching mAbs trough levels—hampers the clinical development of DNA-based mAb therapeutics. Despite its significance for therapeutic applications, little is known about the factors that drive the in vivo expression of mAbs following intramuscular electroporation of mAb-encoding plasmids. Further improvement of in vivo mAb expression requires insight into the fate of the plasmids post-electroporation. 

The current study aims to characterize the in vivo expression of antibodies after pDNA administration on a broader basis than only plasma protein expression. To the best of our knowledge, it is the first study to investigate pDNA levels, mAb mRNA levels, and pDNA localization over an extensive time period after mAb gene transfer. This will lead to a better understanding of the different factors driving protein expression. The findings of this study can contribute to the optimization of DNA-based mAb delivery. On top of that, the techniques described can also be used as a toolbox for evaluating future plasmid constructs or alternative DNA-based delivery methods. 

## 2. Materials and Methods

### 2.1. pDNA Constructs 

Two previously designed pDNA constructs, pCAG-4D5-HC and pCAG-4D5-LC, encoding the heavy chain and light chain, respectively, of the murine anti-HER2 mAb, 4D5 were used [12]. The plasmid backbone of both contained a CAG promoter, pUC origin of replication, ampicillin resistance gene, and TK poly(A) sequence [4]. pDNA was produced in *E. coli* (TOP10F’), purified using the NucleoBond Xtra Maxi EF kit (Machery—Nagel, Düren, Germany), eluted in sterile water, and stored at −20 °C until use. Plasmid purity and concentration were determined via UV spectrophotometry, and integrity was evaluated via agarose gel electrophoresis.

### 2.2. Mice 

Female 7–8-week-old Balb/c mice (Balb/cAnNRj) with an approximate weight of 17–20 g were purchased at Janvier (Janvier, Le Genest-Saint-Isle, France). All experiments were approved by the KU Leuven Animal Ethics Committee (project P157/2017). 

### 2.3. Intramuscular Electroporation 

pDNA delivery to mice was performed as described previously [5]. In short, mice were treated in both tibialis anterior muscles. The site of delivery was prepared using depilatory cream (Veet, Reckitt Benckiser, Slough, UK) approximately one week before treatment. On the day of treatment, mice received an intramuscular injection with 40 µL of 0.4 U/µL hyaluronidase from bovine testes (H4272, Sigma, St. Louis, MO, USA) (reconstituted in sterile saline), exactly one hour prior to pDNA electrotransfer. Intramuscular injection with an equimolar mixture of pCAG-4D5-HC and pCAG-4D5-LC pDNA, 30 µL of 2 µg/µL solution in sterile water, was immediately followed by in situ electroporation using the NEPA21 Electroporator (Sonidel, Dublin, Ireland) with CUY650P5 tweezer electrodes (Sonidel, Dublin, Ireland) at a fixed width of 5 mm. Signa Electrode Gel (Parker Laboratories, Fairfield, NJ) was applied to the muscle and electrodes to decrease tissue impedance below 0.3 kΩ. Three series of four 20-ms square-wave pulses of 120 V/cm with a 50-ms interval were applied with polarity switching after two of the four pulses. Pulse delivery was verified using the NEPA21 readout; if energy transfer was below 0.7 J, an extra pulse was administered.

### 2.4. Sample Collection 

Sample collection was performed at multiple timepoints post treatment (1, 4, 7, 14, 21, 42, and 84 days). At each timepoint, muscle isolation was performed for 5 mice. Blood for mAb titer determination was collected via retro-orbital bleeding, with 0.2 mM sodium citrate buffer as anticoagulant (10% of blood volume), processed to plasma by centrifugation (10 min, 2000× *g*, 4 °C and 10 min, 16,000× *g*, 4 °C), and stored at −20 °C. Mice were euthanized with a Dolethal® (pentobarbital) (Vetoquinol, Niel, Belgium) overdose (150–200 mg/kg) injected peritoneally. Blood samples for anti-drug antibody determination were collected at the end of study follow-up, with a prefilled heparin (Leo Pharma, Ballerup, Denmark) syringe via terminal cardiac puncture, processed to plasma by centrifugation (10 min, 2000× *g*, 4 °C and 10 min, 16,000× *g*, 4 °C) and stored at −20 °C. Tibialis anterior muscles were collected after transcardial perfusion with heparinized PBS. Muscles were snap frozen in isopentane (Sigma, St. Louis, MO, USA), pre-chilled on a dry ice ethanol slurry, and stored at −80 °C until further processing. 

### 2.5. DNA/RNA Extraction from Muscle

DNA and RNA extraction were performed using the AllPrep DNA/RNA/miRNA Universal Kit (Qiagen, Hilden, Germany). Homogenization of muscle tissue was performed in Buffer RLT supplemented with 2-mercaptoethanol (Sigma, St. Louis, MO, USA), using metal bead lysing matrix S (MP Biomedicals, Irvine, CA, USA) and a FastPrep-24 classic homogenizer (6.5 m/s, 45 s) (MP Biomedicals, Irvine, CA, USA). Homogenate corresponding to maximum 30 mg of muscle tissue was used for further processing, according to the manufacturer’s instructions. DNA and RNA samples were stored at −20 °C and −80 °C, respectively, until analysis. 

### 2.6. Nuclei Isolation from Muscle 

The protocol for nuclei isolation from frozen muscle was based on a protocol by Santol et al. [13]. Muscles were thawed on ice for 5 min before adding 0.3 mL of ice-cold lysis buffer (10 mM Tris-HCl (Sigma, St. Louis, MO, USA), 10 mM NaCl (Fisher Scientific, Waltham, MA, USA), 3 mM MgCl_2_ (Sigma, St. Louis, MO, USA), and 0.1% NonidetTM P40 (Alfa Aesar, Haverhill, MA, USA) in nuclease-free H_2_O (Invitrogen, Waltham, MA, USA)). Tissue was cut into small pieces (2–3 mm), 4 extra volumes of lysis buffer were added, and the mixture was incubated for 3 min on ice with gentle shaking. Wash buffer (2.7 mL, PBS + 2% bovine serum albumin (BSA) (Sigma, St. Louis, MO, USA)) was added, and the mixture was Dounce homogenized on ice. The cell suspension was filtered through a 70-µm and 40-µm cell strainer (VWR, Leuven, Belgium). Subsequently, the filtrate was centrifuged for 5 min, 500× *g* at 4 °C. The nuclei pellet was washed three times with wash buffer, followed by centrifugation for 5 min, 500× *g* at 4 °C. 

### 2.7. DNA Extraction from Nuclei

Isolated nuclei were lysed by incubation in 0.6 mL nuclei lysis buffer (0.5% SDS (Sigma, St. Louis, MO, USA), 100 µg/mL proteinase K (Thermo Fisher, Waltham, MA, USA), and 20 µg/mL RNase (Qiagen, Hilden, Germany) in nuclease-free H_2_O) for 5 h at 50 °C. DNA was extracted from lysates by one extraction in one volume of TE-saturated phenol (Sigma, St. Louis, MO, USA), one extraction in one volume of mixture TE-saturated phenol:chloroform (Acros Organics, Waltham, MA, USA) 50:50, and one extraction in one volume of chloroform. Total DNA was precipitated with 3 volumes of 95% ethanol (Fisher Scientific, Waltham, MA, USA) containing 0.12 M sodium acetate (Sigma, St. Louis, MO, USA) overnight at −20 °C, washed with 80% ethanol, and stored in elution buffer (Qiagen, Hilden, Germany) at −20 °C until analysis. 

### 2.8. (RT)-qPCR 

A DNA probe-based multiplex real-time quantitative PCR was designed to simultaneously measure pCAG-4D5-HC and pCAG-4D5-LC in total DNA extracted from muscle samples. The method was validated for target specificity, sensitivity, PCR efficiency, and linearity. Quantitative PCR (qPCR) was performed using a Lightcycler 480 (Roche, Basel, Switzerland). The reaction mix included 2× qPCRBIO Probe Mix (PCR Biosystems, Wayne, PA, USA), primers (final concentration 0.4 µM), and probes (final concentration 0.2 µM) (Table 1). qPCR conditions were as follows: activation at 95 °C for 3 min, denaturation/amplification (40 cycles of 95 °C for 5 s and 62 °C for 30 s). Before data analysis, color compensation was applied to correct for bleed-through of the fluorescence signal. For RNA quantification of 4D5-HC and 4D5-LC, mRNA was reverse-transcribed using 10-fold diluted RNA samples, anchored oligodT primers (Integrated DNA Technologies, Leuven, Belgium), and Superscript II reverse transcriptase (Invitrogen, Waltham, MA, USA), according to the manufacturer’s instructions. cDNA was stored at −20 °C and analyzed with the above-described qPCR method. The 4D5 DNA and mRNA levels are reported as relative values, with the whole muscle data of the first mouse in the 24 h group as the control subject (100%). 

### 2.9. ELISA for mAb Quantification

The 4D5 levels in plasma samples were quantified using a previously described in-house developed ELISA [4], with minor adjustments to the protocol. Briefly, plates were coated overnight at 4 °C with 500 ng/mL human HER2 (10004-H08H, Sino Biologicals, Eschborn, Germany) in PBS. Plates were blocked with 1% BSA in PBS for two hours at room temperature (RT). Samples were diluted in PTAE (PBS 0.1% BSA, 0.002% Tween 80 (Sigma, St. Louis, MO, USA), 5 mM EDTA (Chemlab, Zedelgem, Belgium)) and incubated on the blocked HER2-coated plates for one hour at RT. Serial two-fold dilutions of purified 4D5, with concentrations ranging between 0.125 and 8 ng/mL, were used as the calibration curve. Detection of the captured 4D5 was performed with goat anti-mouse IgG–HRP (GAM IgG (H+L)-HRP, 1:2000 dilution in PTA (PBS 0.1% BSA, 0.002% Tween 80)) (Bio-Rad Laboratories, Hercules, CA, USA). Plates were developed for 30 min using o-phenylenediamine (TCI, Zwijndrecht, Belgium) and H_2_O_2_ (Merck, Rahway, NJ, USA) in citrate (Sigma, St. Louis, MO, USA) buffer. The reaction was stopped with 4 M H_2_SO_4_ (Merck, Rahway, NJ, USA). Absorbance was measured at 490 nm using an ELx808 Absorbance Microplate Reader (BioTek Instruments, Bad Friedrichshall, Germany). Each incubation step was preceded by a washing step with PBS 0.05% Tween 20 (Sigma, St. Louis, MO, USA). Sample concentrations were calculated based on the 4D5 calibration curve using a simple linear regression fit (GraphPad Prism 9.3, GraphPad Software, San Diego, CA, USA).

### 2.10. ELISA for Detection of Anti-Drug Antibodies

The presence of anti-drug antibodies (ADA) against the in vivo expressed mAb was assessed via an affinity capture elution (ACE) ELISA. Ninety-six-well plates were coated overnight at 4 °C with 5 µg/mL purified 4D5 diluted in 1 M sodium bicarbonate (VWR, Leuven, Belgium) buffer, pH 9.6. Plates were blocked with 1% BSA in PBS for one hour at RT, washed, and 25 µL per well of 1 M tris (Sigma, St. Louis, MO, USA) buffer (pH 9.5) was added to the plates. Plasma samples were diluted in PTAE and acidified by 1:10 dilution with 300 mM acetic acid (Honeywell, Charlotte, NC, USA) followed by 15 min incubation at RT. Then, 75 µL of the acidified samples was added to the buffered 4D5-coated plates followed by incubation for one hour at RT to capture any anti-drug antibodies (ADA) present in the samples. Plates were washed, followed by addition of 100 µL 300 mM acetic acid per well and incubation for 15 min at RT to elute bound ADAs. Fresh 96-well plates were loaded with 25 µL 1 M tris buffer (pH 9.5). Then, 75 µL of acid-eluted sample was transferred to the new buffer-containing plate, followed by a one-hour incubation at RT to allow for adsorption of the ADAs to the plate. Plates were blocked with 1% BSA in PBS for one hour at RT. Subsequently, biotinylated 4D5 (diluted to 1 µg/mL in PTA) was added and incubated one hour at RT. Detection was performed with streptavidin poly-HRP (1:5000 dilution in PTA) (Sanquin, Amsterdam, The Netherlands) and incubation for 30 min at RT. Each incubation step, except the two blocking steps, was preceded by a washing step with PBS 0.002% Tween 80. Plates were developed for 60 min using o-phenylenediamine and H_2_O_2_ in citrate buffer. The reaction was stopped with 4 M H_2_SO_4_. Absorbance was measured at 490 nm using an ELx808 Absorbance Microplate Reader (BioTek Instruments, Bad Friedrichshall, Germany). 

### 2.11. DNA Scope 

For the DNA scope assay, three animals were included from three time points post-pDNA electroporation (24 h, week 1 and week 6). Per animal, six longitudinal tissue slices, located in the middle of the tibialis anterior muscle tissue, were evaluated. Snap-frozen muscles were sectioned longitudinally at a thickness of 5 µm using a cryostat and mounted onto glass slides (Superfrost® plus, Fisher Scientific, Waltham, MA, USA). RNAScope Multiplex Fluorescent Reagent Kit v2 (ACD, Advanced Cell Diagnostics, Newark, CA, USA) was used following the manufacturer’s instructions for fresh frozen tissue, except for an extended fixation time (one hour) in 10% neutral buffered formalin (Sigma, St. Louis, MO, USA). Custom-designed probes targeting the variable domain of pCAG-4D5-HC and pCAG-4D5-LC were purchased from ACD (Advanced Cell Diagnostics, Newark, CA, USA). Opal dyes 570 (1:1500) and 690 (1:1000) (Akoya Biosciences, Marlborough, MA, USA) were used for developing the fluorescence signal originating from pCAG-4D5-HC and pCAG-4D5-LC, respectively. Images were collected by a Zeiss LSM 880 confocal microscope using a 40× oil-immersion objective (Zeiss, Oberkochen, Germany). 

The amount of pDNA in the nuclei was quantified relative to the total amount of pDNA. Microscopy scanning was performed on the part of the muscle slices that contained the highest amount of pDNA, selected based on visual inspection. Z-stacks were analyzed in three dimensions using Imaris 9.6.13 software (Oxford Instruments, Abingdon, United Kingdom). Nuclei were detected as surface objects, DNA signal for pCAG-4D5-HC and pCAG-4D5-LC as spots. Colocalization was determined based on shortest distance calculations and expressed as percentage of nuclear spots for each channel separately. 

### 2.12. Statistics

At the start of experiments, mice were randomized based on body weight. DNA/RNA quantification and DNA scope data are available from multiple timepoints. To compare these timepoints, data are analyzed using one-way ANOVA, when data is normally distributed (Shapiro–Wilk test), or Kruskal–Wallis test, when this is not the case, with Tukey’s/Dunn’s multiple comparisons test. The decrease of plasma 4D5 levels from week 3 onwards was analyzed using a linear mixed model in R. Nuclei pDNA quantification data is analyzed with unpaired t-tests, to investigate a possible difference between the two timepoints. Data with *p*-value below 0.05 are considered as statistically significantly different. Statistical analysis and figure drawing were completed using GraphPad Prism 9.3.1 (GraphPad Software, San Diego, CA, USA). All graphs display individual data points per animal or tissue slice and median values or average ± SD (specified in figure legends). 

## 3. Results

### 3.1. pDNA Levels after Electrotransfer in Muscle

To monitor the quantity of pDNA at different time points after intramuscular electroporation, 35 mice were treated with pCAG-4D5-HC and pCAG-4D5-LC followed by electroporation. At seven different time points post treatment, ranging from 24 h to 12 weeks, five animals were sacrificed, and treated muscles were isolated. Negative control samples were muscles isolated from non-injected, non-electroporated mice of the same strain, gender, age, and weight. pDNA was extracted from muscle samples and subjected to qPCR. As a positive control for DNA/RNA extraction efficiency and qPCR, a non-treated muscle was spiked with the pDNA constructs before homogenization. 

Levels of pCAG-4D5-HC and pCAG-4D5-LC are represented relative to the amount of the control subject at 24 h (Figure 1). Levels for the heavy and the light chain showed a highly similar profile during the 12-week follow-up. At day 4, the median pDNA level was around 20% (range: 11.25–218%), further decreasing to less than 10% at week 1. At the final evaluation point, 12 weeks, less than 0.5% of pDNA was still present in all animals compared to the reference animal at 24 h (Figure 1). pCAG-4D5-HC and pCAG-4D5-LC data were normally distributed and subject to one-way ANOVA combined with Dunn’s test for multiple comparisons. A significant difference in mean (i.e. a decrease of pDNA over time) was observed between 24 h and all other timepoints (day 4: *p* = 0.029, other timepoints: *p* < 0.0001). 

### 3.2. mRNA Expression Levels Following pDNA Electrotransfer in Muscle

Similar to pDNA, expression profiles were highly similar between heavy chain and light chain encoding mRNA during the 12-week follow-up. mRNA expression levels showed a high inter-animal variation. However, no significant difference was observed between different time points based on the Kruskal–Wallis test (*p* > 0.05), indicating that the mRNA expression levels are on average stable over time (Figure 2). 

### 3.3. 4D5 Antibody and ADA Plasma Levels

Using a linear mixed model, we could demonstrate that 4D5 plasma levels changed significantly between day 21 (peak level) and day 84 (t(16.3) = −5.13, *p* = 9.43e^−05^), with significant changes when comparing to day 21 for both day 42 (t(15) = −2.43, *p* = 0.0273) and day 84 (t(15) = −5.585, *p* = 4.14e^−05^). Plasma 4D5 concentrations showed a steady increase up to day 14, after which average levels remained around 4 µg/mL up to day 21. The 4D5 concentrations decreased over the course of the next 9 weeks. At the end of the follow-up period, 12 weeks post treatment, average levels were around 2 µg/mL, approximately half of the peak levels (Figure 3). For none of the animals, ADAs were detected during the 12-week follow-up. 

### 3.4. Fate of pDNA after Intramuscular Electroporation

The fast decrease of pDNA seems to be in contradiction with the stable mRNA levels and the long-term mAb expression. Therefore, it is hypothesized that only a small fraction of the injected pDNA reaches the nuclei where it can be transcribed into mRNA followed by translation to protein while the majority of the pDNA remains outside the nuclei and gets degraded over time without contributing to the protein expression. To verify this hypothesis, pDNA localization was performed at different time points following intramuscular administration and electroporation (i.e., 24 h, week 1 and week 6). 

Figure 4 shows representative examples of processed images for the three different time points. Remarkable is that DNA-containing nuclei often have more than one pDNA copy, but that the majority of the nuclei does not contain any pDNA copy. 

Nuclear pDNA is represented as percentage of total spots located in the nuclei (Figure 5A). For pCAG-4D5-HC, data were normally distributed and subject to one-way ANOVA combined with Tukey’s test for multiple comparisons. A significant difference in mean (i.e., an increased degree of nuclear pDNA over time) was observed between 24 h and 6 weeks (*p* = 0.0001) and between week 1 and week 6 (*p* < 0.01). There was no statistically significant difference between 24 h and week 1 (*p* > 0.05). For pCAG-4D5-LC, data were not normally distributed for all time points (Shapiro-Wilk test) and were subjected to a non-parametric Kruskal–Wallis test followed by the Dunn’s multiple comparisons test. A statistically significant difference in sum of ranks (i.e., an increased degree of nuclear pDNA over time) between 24 h and 6 weeks (*p* < 0.02) was observed. Figure 5B shows the absolute number of counted dots in all tissue slices, grouped per time point. Kruskal–Wallis with Dunn’s multiple comparisons test showed a statistically significant difference (i.e., a decrease over time) between 24 h and week 1 (*p* < 0.005), between 24 h and week 6 (*p* < 0.0001), and between week 1 and week 6 (*p* < 0.05). 

### 3.5. Nuclear pDNA Levels after Electrotransfer 

To substantiate the DNA scope results, nuclei were isolated and intranuclear pDNA of muscles was quantified at day 4 and week 2. pCAG-4D5-HC and pCAG-4D5-LC were quantified separately and showed similar results. At day 4, the median intranuclear pDNA level was around 0.5% (range: 0.08–3.65%) of the whole muscle pDNA of the control subject at 24 h, further increasing slightly to 1% at week 2 (range: 0.69–7.99%) (Figure 6). The intranuclear pDNA levels were not significantly different between these two timepoints (Mann–Whitney U test, *p* > 0.05). This is in contrast with the pDNA quantification in the whole muscle, where a clear decrease of pDNA was observed between day 4 and week 2 (Figure 1, early time points repeated in Figure 6). 

This nuclear pDNA quantification allows to calculate percentage of nuclear pDNA based on whole-muscle data instead of region-specific data from the DNA scope assay. Percentage of nuclear pDNA is calculated compared to average whole-muscle pDNA levels. The same evolution was observed as with DNA scope, i.e., the percentage of nuclear pDNA increases significantly over time. For both pCAG-4D5-HC and pCAG-4D5-LC, a Mann–Whitney U test revealed a statistically significant difference in percentage of nuclear pDNA between 4 days and 2 weeks (*p* = 0.0159), increasing from around 1% at day 4 to around 35% at week 2 (Figure 7). 

## 4. Discussion

Antibody gene transfer is a promising alternative for conventional antibody treatment. Despite being under clinical evaluation, little is currently known about the fate of the injected pDNA after intramuscular electroporation. In the context of pDNA vaccination, pDNA biodistribution following intramuscular electroporation is well-characterized both pre-clinically and clinically [14,15,16]. Following intramuscular injection and electroporation, pDNA is transiently distributed through the body, but only persists at the site of injection. Moreover, no obvious risk of integration into the host genome has been reported in vaccination studies. pDNA is typically lost within a few months, likely due to plasmid loss consequent to cell turnover, plasmid silencing, and/or immune responses, including cellular immunity [14,15,16]. It is currently unclear how these factors play a role in the context of antibody gene transfer, where, in contrast to vaccination, the absence of an immune response is pursued, both against the plasmid and the expressed transgene. Despite the large number of pre-clinical studies for DNA-based mAbs, there is little to no data available on the factors that impact transgene expression. 

This study characterizes antibody gene transfer from a fundamental point of view. DNA, the starting point of the central dogma of molecular biology, is injected as part of the treatment. In muscle, pDNA levels decreased 90% between 24 h and one week post treatment. In contrast, mRNA levels remained stable over time. 4D5 antibody plasma concentrations reached peak levels at week two followed by a slow decrease (5% after 12 weeks). Evaluation of pDNA localization revealed that extranuclear pDNA was cleared fast, whereas the nuclear fraction remained relatively stable. This is in line with the observed mRNA and protein levels over time and indicates that only a minor fraction of the administered pDNA is ultimately responsible for the observed systemic mAb levels. 

A previous study by Cappelletti et al. [17] showed that within 48 h post injection, the pDNA levels decreased 10,000-fold compared to the injected dose. No pDNA quantification was performed at later time points. In the current study, the initial pDNA loss (within 24 h) appears to be similar to the pDNA loss within 48 h in the study by Cappelletti et al. [17]. However, more interesting is the amount of pDNA that remains over a longer time period. We therefore evaluated the relative pDNA quantity starting from 24 h post treatment up to 12 weeks. This pDNA time profile shows a substantial decrease in pDNA not correlating with evolution of the transgene expression, implying that even a small amount of remaining pDNA is enough to result in substantial protein expression. A similar observation was reported by Molnar et al. [18]. The main difference in study design was the use of SCID mice and a plasmid coding for the reporter gene β-galactosidase instead of a therapeutic antibody [18]. pDNA levels decreased 50% between 10 and 90 days, a smaller decrease than observed in the current study, 90% pDNA decrease between week 2 and week 12. Possible explanations are the differences in study design and different sample processing, whole muscle pDNA extraction in this study versus pDNA extraction of 10 µM thick muscle cryosections in the study from Molnar et al. [18].

Muscle tissue is terminally differentiated, and normal rodent adult muscle has a slow turnover rate, at most 1–2% of myonuclei are replaced per week [19]. Therefore, this cannot explain the substantial loss of pDNA. Other possible reasons for the pDNA decline could be apoptosis of transfected cells or the lack of stability of extrachromosomal pDNA [18]. A possible disadvantage of using qPCR for pDNA quantification is that not only intact but also degraded pDNA can be detected because only part of the pDNA is amplified during qPCR, resulting in a positive signal. However, data from the literature show that pDNA degradation is mainly a concern in the first hours post treatment and should not have a major influence on pDNA quantification at later time points [17]. 

To elucidate the nature of the apparent contradiction (strong decrease of pDNA whereas mRNA and protein levels remain relatively stable over the time course studied), pDNA in slices of treated mouse muscle (24 h, 1 week, and 6 weeks) was visualized and quantified using DNA scope. An important consideration is that these data are based on the analysis of only a fraction of the treated muscle (region with highest pDNA content in six muscle sections (5 µm) per muscle). The DNA scope results confirm a pDNA decrease as observed with qPCR. On the other hand, the results show a significant increase in the percentage nuclear pDNA over time. This most probably indicates that the pDNA that disappears is mainly extranuclear pDNA, either extracellular or cytoplasmic. This finding was confirmed by quantifying nuclear pDNA 4 days and 2 weeks post treatment, indicating a stable pDNA level in nuclei, and as a consequence, an increasing percentage of nuclear pDNA over time. Both extracellular and cytoplasmic pDNA can be removed by nucleases. It is known that pDNA in cytoplasm has a limited stability, as shown by in vitro studies in which pDNA half-life in cytoplasm was determined to be only 90 min [20,21]. 

Overall, this study provides evidence that the efficiency of pDNA reaching the nucleus is rather low. Previous research was especially focused on facilitating DNA entry in the cell [22]. However, to result in protein expression, pDNA needs to overcome three main obstacles, (1) the plasma membrane, (2) cytoplasm, and (3) the nuclear membrane [21]. Crossing the first barrier is mediated by electroporation, which results in a temporal poration of the cell membrane, allowing the pDNA to enter the cell. The pDNA that reaches the cytoplasm ends up in a dense network of organelles and cytoskeletal elements, preventing the diffusion of large molecules. pDNA is assisted by the microtubule network and associated motor proteins to traffic towards the nucleus [23]. Nevertheless, it is assumed that a lot of naked pDNA will be degraded in the cytoplasm by cytosolic nuclease(s) [20]. The final goal is pDNA reaching the nucleus, where it can be transcribed into mRNA, which can then be translated into protein. Research to overcome this final hurdle is gaining attention as this seems to be the rate-limiting step for gene transfer in non-dividing cells [22].

pDNA can enter the nucleus through the nuclear pore complex (NPC). As a consequence, enhanced transfection ability is observed in mitotic cells displaying a disassembled nuclear envelope [20]. However, antibody gene transfer often targets tissues consisting of non-dividing cells, which further adds to the difficulty of pDNA entering the nucleus. pDNA molecules associate with polypeptides such as transcription factors that contain nuclear localization sequences (NLS), which consequently interact with nuclear import receptors [21]. Strategies are being explored to facilitate nuclear translocation of pDNA [24]. As NLS play a critical role in nuclear uptake, modification of plasmids with NLS peptides or covalent linking of the nuclear targeting sequences to the pDNA constructs could improve nuclear uptake [23]. Other possibilities that have been explored are (1) coupling of the pDNA to recombinant importin β, which can bind to NPC [25]; (2) attaching dexamethasone to plasmids, which could result in glucocorticoid receptor binding and could consequently lead to conformational changes of the receptor exposing an NLS [26]; or (3) chemical modification of pDNA vectors with oligosaccharides, which can bind to the lectins in the NPC [27]. All these strategies have a major impact on the complexity of the production process of these constructs and are therefore not viable options as they overturn one of the main advantages of antibody gene transfer, i.e., rapid design and production at large scale, saving both time and resources.

On the other hand, nuclear import of pDNA is a sequence-specific process. Even pDNA without NLS or other conjugates as described above is able to enter the nucleus to a certain extent. Sequence optimization can lead to increased pDNA uptake in the nucleus [23]. Previous research has shown that when the SV40 promoter/enhancer region is inserted in a plasmid that cannot enter the nucleus, nuclear import was observed. A total of 72 bp of the SV40 enhancer, referred to as DNA nuclear targeting sequence (DTS), have shown to be sufficient to drive plasmid nuclear import in different cell lines [23]. A limited number of in vivo studies has evaluated this improved nuclear uptake by adding this DTS to the pDNA, resulting in a 40–200-fold increase in gene expression for vascular gene transfer in rats [28], a 16-fold higher luciferase reporter expression after intramuscular electroporation in mice [29], and a 20-fold increase in the level of exogenous gene expression in muscle [30]. However, the studies evaluating intramuscular gene transfer only evaluated the levels of the expressed transgene and did not report data at the level of the plasmid. The DNA scope data from this study endorse the need for improving nuclear pDNA uptake. In addition to the low percentage of nuclear pDNA, data show that only few nuclei contain pDNA, often multiple copies, and that the majority of the nuclei do not contain any pDNA. Possibly, these few well-transfected nuclei had a damaged nuclear envelope due to electroporation or mitosis. 

In addition to an elaborate investigation of the fate of pDNA after antibody gene transfer, this study measured the mRNA and protein levels in the same subjects. mRNA levels show huge variability, but no significant difference is observed between different time points. This contributes to the hypothesis that the amount of pDNA in the nuclei stays quite stable over time. The pharmacokinetic profile of the protein expression shows an initial increase in antibody titers up to week 3, followed by a steady decline in plasma protein levels. 

Unfortunately, the data of this study do not allow to directly correlate the protein expression levels to pDNA and mRNA levels because of several factors, (1) mice are treated in both tibialis anterior muscles: only one is used for pDNA and mRNA quantification, while both contribute to protein expression; (2) isolating muscles for pDNA and mRNA extraction is a terminal procedure, thus no further plasma protein level follow-up in the same animals is possible. Nevertheless, it is likely that pDNA/mRNA levels correlate with plasma protein levels at a later timepoint given the plasma half-life of antibodies. The percentage of nuclear pDNA and the number of pDNA dots of the DNA scope assay do not show a correlation with protein expression either. However, as pointed out before, this data was extracted from one small part of one of two treated muscles, so this experiment cannot exclude the possible correlation either.

In conclusion, this study demonstrates that durable expression is dependent on the nuclear uptake of the pDNA and that efforts to increase the protein levels upon pDNA-based gene therapy should also focus on strategies to increase both cellular entry and migration of the pDNA into the nucleus. The generated toolbox can be used to guide the design and evaluation of future pDNA constructs or delivery methods, to achieve a robust and prolonged protein expression.

## Figures and Tables

**Figure 1 pharmaceutics-15-01160-f001:**
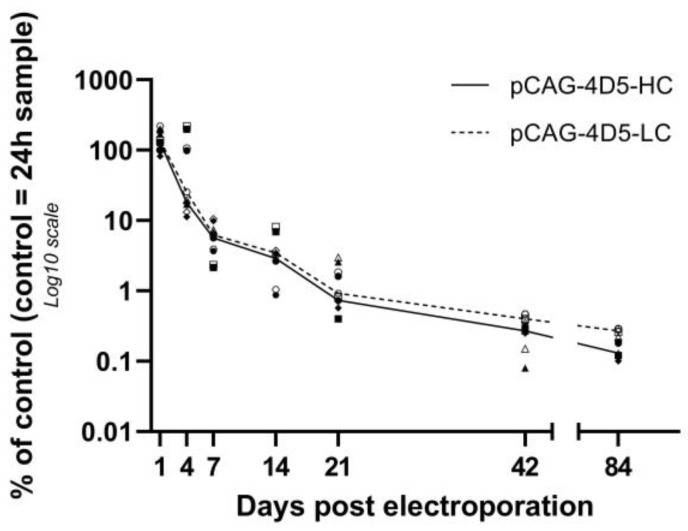
pDNA quantification. Balb/c mice received intramuscular pDNA electrotransfer of 60 µg pDNA (equimolar mixture pCAG-4D5-HC and pCAG-4D5-LC) in the tibialis anterior muscle. Muscles were isolated at multiple time points post treatment, DNA extracted and quantified using qPCR. pDNA quantification (closed symbols: pCAG-4D5-HC, open symbols: pCAG-4D5-LC) is represented relative to the amount of the control subject at 24 h. Data are presented as individual data points per animal (mean of two independent runs, in which samples were run in duplicate). pCAG-4D5-HC and pCAG-4D5-LC of the same animal are indicated with the same symbol. Data from each time point originate from different animals. Curves represent the median per time point (line: pCAG-4D5-HC, dashed line: pCAG-4D5-LC) (*n* = 5, five mice per time point).

**Figure 2 pharmaceutics-15-01160-f002:**
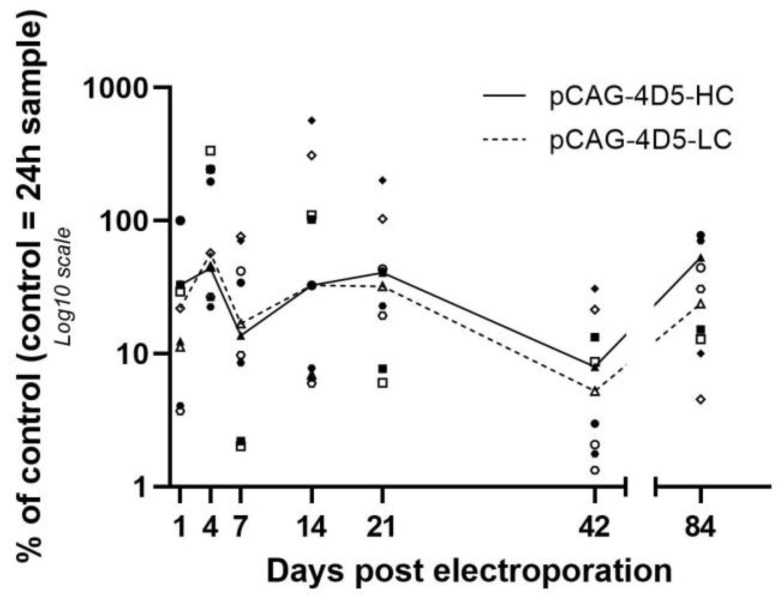
mRNA expression levels. Balb/c mice received intramuscular pDNA electrotransfer of 60 µg pDNA (equimolar mixture pCAG-4D5-HC and pCAG-4D5-LC) in the tibialis anterior muscle. Muscles were isolated at multiple time points post treatment, and mRNA was extracted and quantified using reverse transcription qPCR. mRNA quantification (closed symbols: pCAG-4D5-HC, open symbols: pCAG-4D5-LC) was performed relative to the amount of the control subject at 24 h. Data are represented as individual data points per animal (average of three independent runs, in which samples were run in duplicate). pCAG-4D5-HC and pCAG-4D5-LC of the same animal are indicated with the same symbol. Data from each time point originate from different animals. Curves represent the median per time point (line: pCAG-4D5-HC, dashed line: pCAG-4D5-LC) (*n* = 5, five mice per time point).

**Figure 3 pharmaceutics-15-01160-f003:**
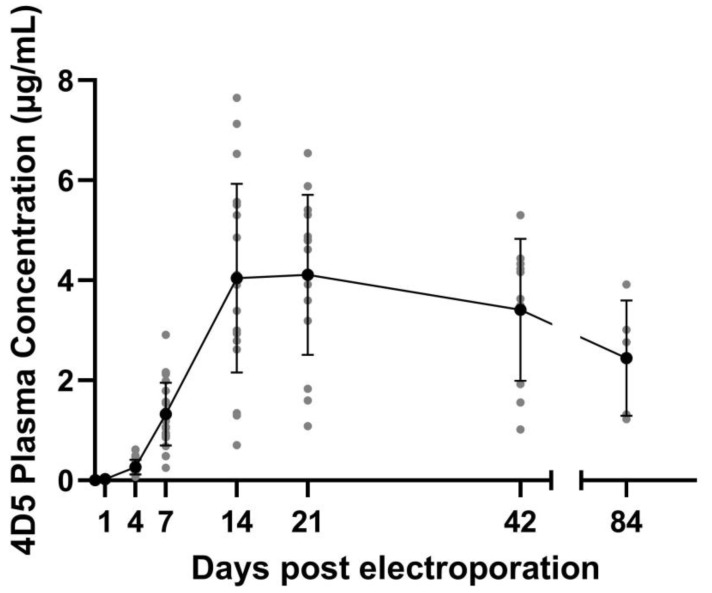
4D5 antibody expression levels. Balb/c mice received intramuscular pDNA electrotransfer of 60 µg pDNA (equimolar mixture pCAG-4D5-HC and pCAG-4D5-LC) in both tibialis anterior muscles. 4D5 mAb plasma concentrations, dots indicate the individual expression levels per animal, the solid line indicates the average levels including standard deviation. Number of animals decreases at each subsequent time point due to the terminal procedure upon muscle isolation (*n* = 35 (24 h, only 5 above LOQ); *n* = 30 (4 days); *n* = 25 (week 1); *n* = 20 (week 2); *n* = 15 (week 3); *n* = 10 (week 6); *n* = 5 (week 12)).

**Figure 4 pharmaceutics-15-01160-f004:**
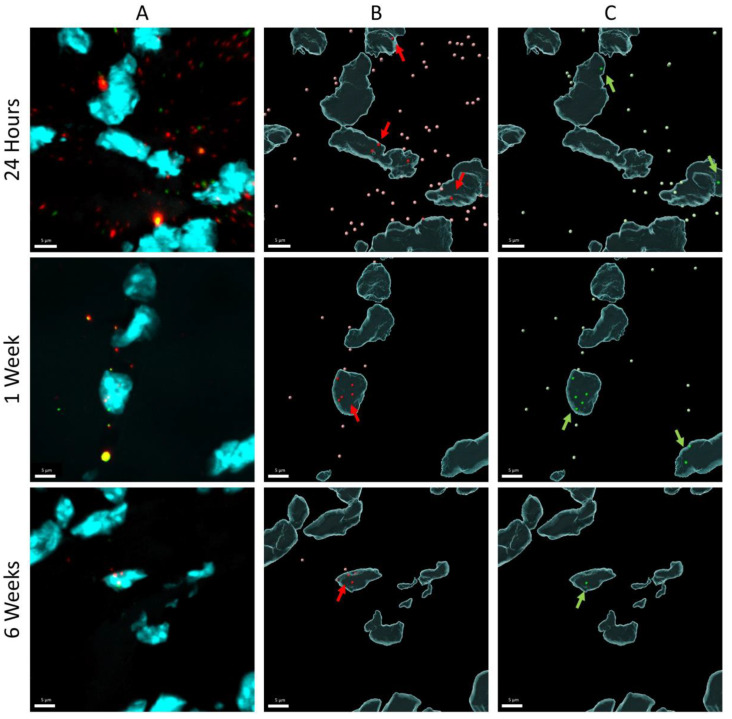
Representative DNA scope Image Processing. Balb/c mice received intramuscular pDNA electrotransfer of 60 µg pDNA (equimolar mixture pCAG-4D5-HC and pCAG-4D5-LC) in the tibialis anterior muscle. DNA scope assay on 5-µm muscle slices, visualized with confocal microscopy for 3 different time points (24 h, 1 week, and 6 weeks). (**A**) Confocal microscopy signal (three channels: (i) DAPI nuclei signal cyan, (ii) pCAG-4D5-HC Opal 570 signal red, (iii) pCAG-4D5-LC Opal 690 signal green). (**B**,**C**) surfaces and spots generated with Imaris software for each channel separately (**B**) spots pCAG-4D5-HC pink (extranuclear)/red (nuclear), (**C**) spots pCAG-4D5-LC light green (extranuclear)/green (nuclear). Arrows indicate regions with nuclear pDNA.

**Figure 5 pharmaceutics-15-01160-f005:**
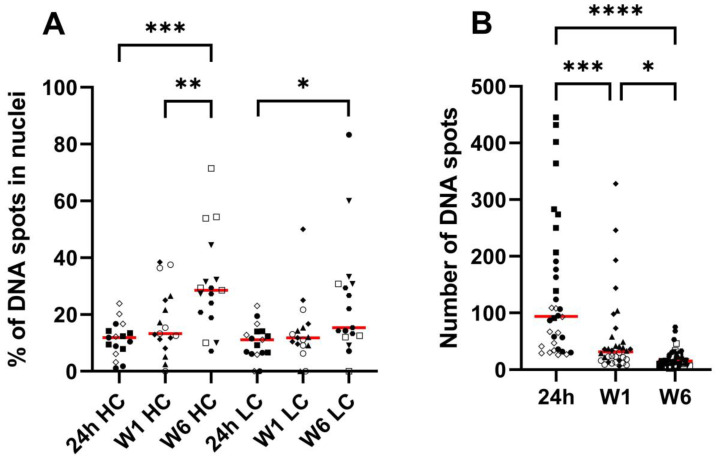
DNA scope pDNA localization and total spot count. Balb/c mice received intramuscular pDNA electrotransfer of 60 µg pDNA (equimolar mixture pCAG-4D5-HC and pCAG-4D5-LC) in the tibialis anterior muscle. (**A**) DNA scope nuclear pDNA, data points indicate the percentage of DNA dots in nuclei in a single tissue slice, data points from tissue slices originating from the same muscle are indicated with the same symbol (3 animals per time point). (**B**) DNA scope DNA spot numbers, data points indicate the amount of DNA spots in a single tissue slice, data points from tissue slices originating from the same muscle are indicated with the same symbol (3 animals per time point). The red lines represent the median per subgroup ((**A**): *n* = 17, (**B**): *n* = 34). Asterisks represent the level of statistical significance: * *p* ≤ 0.05, ** *p* ≤ 0.01, *** *p* ≤ 0.001, **** *p* ≤ 0.0001.

**Figure 6 pharmaceutics-15-01160-f006:**
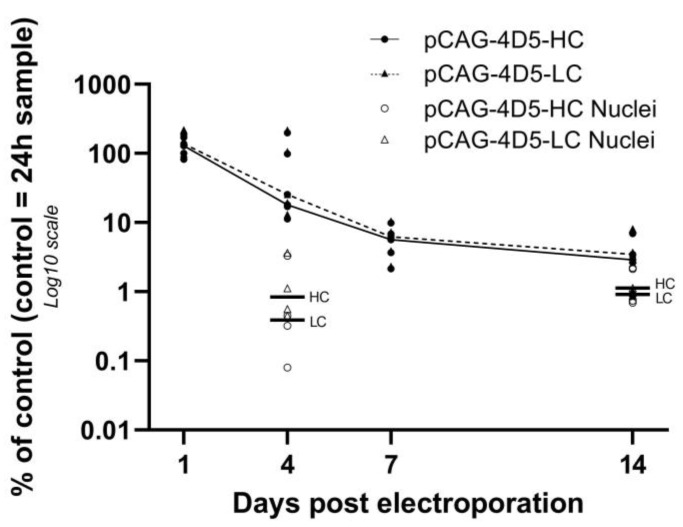
(Intranuclear) pDNA quantification. Balb/c mice received intramuscular pDNA electrotransfer of 60 µg pDNA (equimolar mixture pCAG-4D5-HC and pCAG-4D5-LC) in the tibialis anterior muscle. Muscles were isolated at multiple time points, nuclei were isolated, and DNA was extracted and quantified using qPCR. pDNA quantification (
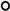

: intranuclear pCAG-4D5-HC, 
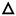
: intranuclear pCAG-4D5-LC, 
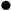
: pCAG-4D5-HC whole muscle, 
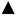
: pCAG-4D5-LC whole muscle) was performed relative to the amount of the control subject at 24 h. Data are represented as individual data points per animal (average of triplicate). Curves represent the median per time point (line: pCAG-4D5-HC, dashed line: pCAG-4D5-LC) (*n* = 5, five mice per time point) for whole-muscle pDNA quantification, horizontal lines represent median intranuclear pDNA levels (day 4: *n* = 4; week 2: *n* = 5).

**Figure 7 pharmaceutics-15-01160-f007:**
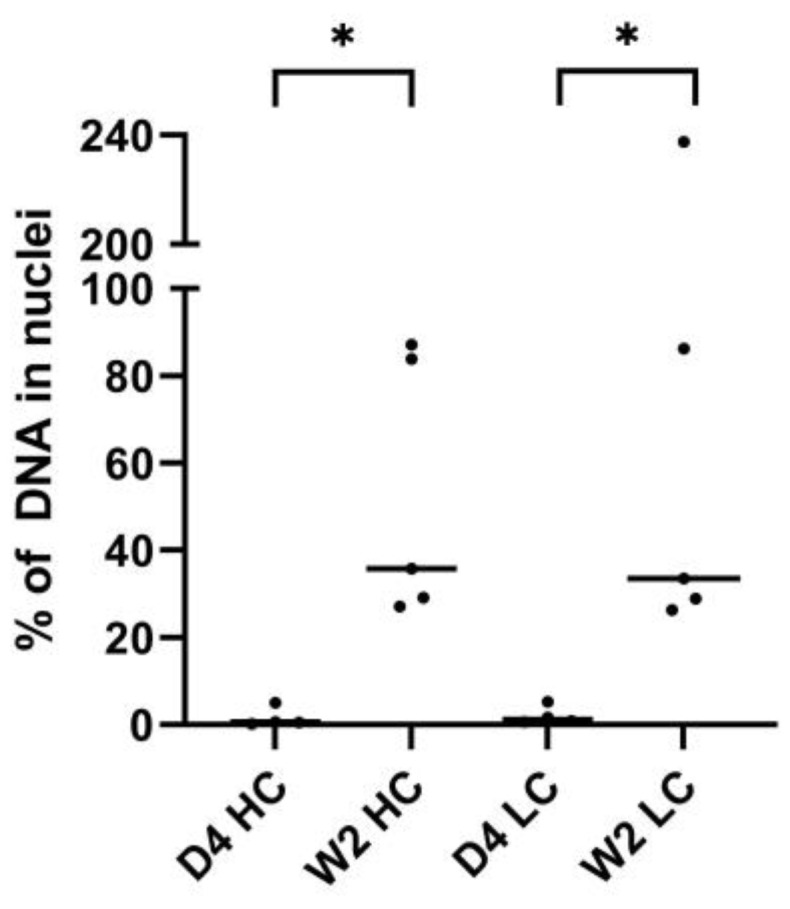
qPCR percentage nuclear pDNA. Balb/c mice received intramuscular pDNA electrotransfer of 60 µg pDNA (equimolar mixture pCAG-4D5-HC and pCAG-4D5-LC) in the tibialis anterior muscle. Nuclear pDNA quantification by qPCR, data points indicate the percentage of total pDNA in nuclei. The horizontal lines represent the median per subgroup (day 4: *n* = 4, week 2: *n* = 5). Asterisks represent the level of statistical significance: * *p* ≤ 0.05.

**Table 1 pharmaceutics-15-01160-t001:** Primer and probe sequences qPCR assay pCAG-4D5-HC and pCAG-4D5-LC purchased from Integrated DNA Technologies (IDT, Leuven, Belgium). Specific binding regions are indicated between brackets.

Oligo Name	pCAG-4D5-HC	pCAG-4D5-LC
Forward Primer	*5’-CCCACCAACGGATACAC (CDR2)*	*5’-AGCCAGGACGTGAACAC (CDR3)*
Reverse Primer	*5’-AGTCCATGGCGTAGAAG (CDR1)*	*5’-TGGGAGGGGTGGTATAG (CDR3)*
Probe	*5’-SUN-TACCTCCTC-ZEN-CAACACAGCCTACCT-3IABkFQ*	*5’-FAM-ACCGACTTC-ZEN-ACCTTCACAATCAGCA-3IABkFQ*

## Data Availability

The data presented in this study are openly available in KU Leuven Research Data Repository (RDR) at 10.48804/CCEHMP [31].

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
