# Peer review of "Exploring the Fate of Antibody-Encoding pDNA after Intramuscular Electroporation in Mice"

_pharmaceutics, 2023, doi:10.3390/pharmaceutics15041160_

Round 1

Reviewer 1 Report

Cuypers et al in their manuscript submitted to Pharmaceutics address important issue of identifying pDNA fate in the muscle cell in relation to antibody presence in the blood stream following in vivo electroporation. Authors have evaluated quantitatively systemic protein production and determined localization of pDNA coding murine anti-HER2 4D5 mAb. Muscle biopsies and blood samples were sampled up to 3 months. Their main observation is that only small fraction of pDNA enters the cell nucleus – an thus only small fraction of pDNA entering the cell contributes to mAb production. Their results are important as they bring to attention that low level of pDNA into cells from extracellular compartment is further reduced by low nuclear uptake of pDNA.

Some points to be further elucidated/explained before publication:

Introduction, page 2, end of 2nd paragraph: The authors raise an interesting point here: “For a broad implementation of DNA-based mAb delivery…” In the previous sentence, the authors state that double-digit ug/ml conc. of mAbs is achieved in mice. Are the therapeutic levels of 5-10 ug/ml mentioned here for humans? It would be nice if authors could comment on the serum mAb levels reached in different species by EP - from small animals to large animals (to humans?). How could one translate the values of serum mAb conc. levels in one species (say mice of double-digit ug/ml) to another specie (humans - 5-10 ug/ml)?

Introduction, page 2, beginning of 3rd paragraph: The research gap or the motivation for this work is not clear/explicit in the introduction. Why a double-digit ug/ml serum mAb concentration in mice is qualified as limited? How much serum mAb concentration in mice will be enough to achieve 5-10 ug/ml serum concentration required in humans? If this part is made explicit, then I belive the research gap or the motivation for this work would be stronger and explicit.

Introduction, page 2, end of 3rd paragraph: Post administration should be clearly defined. Post administration could imply post injection and before electroporation. However, according to the experiments, intramuscular injection of plasmids was 'immediately' followed by in situ electroporation. So, a more appropriate word should be post electroporation.

Materials and Methods, page 3, end of 1st paragraph in which Intramascular electroporation id described. Authors say: “ Pulse delivery was verified using NEPA21 readout, if energy transfer was below 0.7 J and extra pulse was administered.” How is energy related to quality (I assume) of pulse delivery; how 0.7 J value was established as the value requiring additional pulse delivery. How often, in how many animals these occurred and additional pulse was delivered. Were these animals/subpopulation checked later in data analysis for any potential deviation from conventional 4 pulses delivery. 

Results, page 6, 1st paragraph of Results section: If 30 mice were treated, and at 7 different time points, 5 mice were sacrificed, that makes a total of 35 mice. Could the authors clarify, or elaborate, how many mice were treated and how data points correspond to mice?

Later, in the caption of Figure 3, the authors mention "Number of animals decreases at each subsequent time point .... (n=35 (24 h, only 5 above LOQ);..." which indicates that there were 35 mice for the 24 hr time point.

Could the authors clarify this discrepancy?

Results, page 6, end of 1st paragraph of Results section: I understood positive and negative controls here, but what about a control in which pDNA was injected and not electroporated?  This is an important control that is missing from the study. A control like this can help distinguish between extracellular and intracellular DNA. Can the authors explicitly state, or perhaps make it more clear, what the control is over here. 

Results, page 6, beginning of 2nd paragraph of Results section: Is it the levels of plasmid DNA at 24 hrs afters electroporation? If that is so, then what are the values of the positive control and the values of pDNA from 24 hrs to 84 days with respect to the positive control?

In the disucssion, the authors state that - "In the current study, the initial pDNA loss (within 24 hours) was similar to the pDNA loss within 48 hours in the study by Cappelletti et al. [17]." which is about a 10000 fold decrease. There is no data shown to support this claim. Was the initial 10000-fold pDNA loss at 24 hours evaluated using positive control?

(I believe it would not be worthwile to evaluate levels of pDNA in negative control as there should be no DNA to amplify using qPCR).

Page 8, last line of the text just before Figure 4: "major part of the nuclei" or majority of the nuclei?

Page 9, Figure 5: It might be helpful to clarify what the symbols mean in the figure (square, diamond, circle..).

It is important to give the absolute no. of DNA spots in the nuclei. Will it be constant over time as is shown for quantified pDNA in the nucleii relative to the control at 24 h in Figure 6? The relation/percentage given now can be interpreted by the decrease pf DNA spots in the cytosol being larger than in the nuclei; but it can also be a result of increasing (in absolute terms) of DNA spots in nuclei. Which probably is not the case, but authors need to provide absolute values in order to avoid confusion and potential misinterpretation.

Further, it might also be useful to give the total no. of nucleii containing DNA as as percentage of total no. of nucleii for 24 h, W1 and W6. This will give some estimates of the no. of cells containing pDNA to express mAb compared to total no. of cells. (E.g. Less than x% of the cells express the mAb gene. The assumption here is that the whole muscle was exposed to the electric field.) 

Discussion: Few things missing from the discussion are: -

1. mRNA levels showed no significance decrease in 12-week follow-up, whereas plasma mAB concentrations increased up to day 14, remained constant upto day 21 and then decreased over the next 9 weeks. How do the authors explain this? Why is there a decrease in plasma conc. levels when the mRNA levels remain more or less constant.

2. The authors claim that pDNA level in the nucleii remain stable based on observations from day 4 and day 14. Additional information can be obtained from DNA scope if authors can determine the absolute no. of pDNA spots in nucleii on day 1, w1 and w6. This will perhaps help clarify the fate of pDNA in nucleii till w6.

Can the authors comment on the fate of pDNA in nucleii. Does the amount of nuclear pDNA decrease over time (beyond 14)? How does this correspond to stable mRNA levels? And how does this correspond to decreasing plasma mAB levels beyond day 21?

Discussion, page 11, 2nd paragraph: authors state: “In the current study, the initial loss (within 24 hours) was similar to the pDNA loss within 48 hours in the study by Cappeletti et al” - Where is the data to support this claim? Was the initial 10000-fold pDNA loss at 24 hours evaluated using positive control?

Author Response

Answers to the reviewer's questions are provided below in bold. If changes were made in the manuscript, the adjusted text and line numbers are provided. In the revised manuscript, ‘track-changes’ was used to facilitate the review process.

Cuypers et al in their manuscript submitted to Pharmaceutics address important issue of identifying pDNA fate in the muscle cell in relation to antibody presence in the blood stream following in vivo electroporation. Authors have evaluated quantitatively systemic protein production and determined localization of pDNA coding murine anti-HER2 4D5 mAb. Muscle biopsies and blood samples were sampled up to 3 months. Their main observation is that only small fraction of pDNA enters the cell nucleus – an thus only small fraction of pDNA entering the cell contributes to mAb production. Their results are important as they bring to attention that low level of pDNA into cells from extracellular compartment is further reduced by low nuclear uptake of pDNA.

Some points to be further elucidated/explained before publication:

Introduction, page 2, end of 2nd paragraph: The authors raise an interesting point here: “For a broad implementation of DNA-based mAb delivery…” In the previous sentence, the authors state that double-digit ug/ml conc. of mAbs is achieved in mice. Are the therapeutic levels of 5-10 ug/ml mentioned here for humans? It would be nice if authors could comment on the serum mAb levels reached in different species by EP - from small animals to large animals (to humans?). How could one translate the values of serum mAb conc. levels in one species (say mice of double-digit ug/ml) to another specie (humans - 5-10 ug/ml)?

The therapeutic trough levels mentioned are indeed for humans, which is now specified in the introduction. It was a good suggestion to add some data of larger animals to support the clinical translatability of our findings in mice. Previous experiments in sheep have shown similar mAb expression levels. At the highest evaluated dose of an ovine antibody in sheep, mAb levels in plasma were around 5µg/mL. These data are now included in the introduction (Line 65-69):

This is comparable to the levels observed in sheep, where peak plasma levels reached 5 µg/mL after administration of 12 mg pOVAC, in the absence of anti-drug antibodies. Sheep are more similar to humans in terms of body weight, musculature and blood volume and therefore these data provide valuable insights towards clinical translation [7].

Introduction, page 2, beginning of 3rd paragraph: The research gap or the motivation for this work is not clear/explicit in the introduction. Why a double-digit ug/ml serum mAb concentration in mice is qualified as limited? How much serum mAb concentration in mice will be enough to achieve 5-10 ug/ml serum concentration required in humans? If this part is made explicit, then I belive the research gap or the motivation for this work would be stronger and explicit.

The double-digit µg/mL plasma concentration is qualified as limited because peak levels are in line with trough levels of most therapeutic antibodies. However, this is only 2-3 weeks post treatment. After that, levels decrease below trough levels. Ideally these levels need to be maintained above or around the trough value for a longer period of time. To make this more explicit, the text in the introduction has been adapted (Line 74-75):

“The limited mAb expression that can be attained in vivo - only peak levels reaching mAbs trough levels - hampers the clinical development of DNA-based mAb therapeutics.”

Introduction, page 2, end of 3rd paragraph: Post administration should be clearly defined. Post administration could imply post injection and before electroporation. However, according to the experiments, intramuscular injection of plasmids was 'immediately' followed by in situ electroporation. So, a more appropriate word should be post electroporation.

Thank you for this suggestion. The wording is now changed to the suggestion, “post electroporation”. (Line 79)

Materials and Methods, page 3, end of 1st paragraph in which Intramascular electroporation id described. Authors say: “Pulse delivery was verified using NEPA21 readout, if energy transfer was below 0.7 J and extra pulse was administered.” How is energy related to quality (I assume) of pulse delivery; how 0.7 J value was established as the value requiring additional pulse delivery. How often, in how many animals these occurred and additional pulse was delivered. Were these animals/subpopulation checked later in data analysis for any potential deviation from conventional 4 pulses delivery.

The energy value is directly correlated with the resistance. If the energy transfer is below 0.7 J, the resistance was too high. This can be due to insufficient application of Electrode gel, bad placement of the electrodes, … The cut-off value of 0.7 was previously determined based on in house experiments for optimization of the electroporation protocol. In the current study an extra series of pulses was applied for 14 out of 70 treated muscles (20 %). Data sets were checked but no deviation was observed in the animals that received an extra series of pulses.

Results, page 6, 1st paragraph of Results section: If 30 mice were treated, and at 7 different time points, 5 mice were sacrificed, that makes a total of 35 mice. Could the authors clarify, or elaborate, how many mice were treated and how data points correspond to mice?

Later, in the caption of Figure 3, the authors mention "Number of animals decreases at each subsequent time point .... (n=35 (24 h, only 5 above LOQ);..." which indicates that there were 35 mice for the 24 hr time point.

Could the authors clarify this discrepancy?

Thank you for pointing out this error. 35 mice were used in the experiments, as you deduced from the fact that there are 7 timepoints and 5 mice are sacrificed per time point. This mistake is now corrected. (Line 268)

Results, page 6, end of 1st paragraph of Results section: I understood positive and negative controls here, but what about a control in which pDNA was injected and not electroporated?  This is an important control that is missing from the study. A control like this can help distinguish between extracellular and intracellular DNA. Can the authors explicitly state, or perhaps make it more clear, what the control is over here.

In this study, no positive control consisting of pDNA injection without electroporation was included. We only distinguish between intranuclear and extranuclear pDNA and not between intracellular and extracellular pDNA. This could be an interesting question to address in follow-up studies. However, with the currently applied methodology, we are not able to investigate the difference in intracellular and extracellular pDNA.

The positive control mentioned on page 6 is only used as a control for DNA/RNA extraction efficiency and qPCR. A known amount of pDNA was spiked to a muscle of a non-treated animal before homogenization. To clarify that the positive control was only a control for DNA/RNA extraction and qPCR, and not for the treatment, the text is adapted accordingly (Line 272-274):

“pDNA was extracted from muscle samples and subjected to qPCR. As a positive control for DNA/RNA extraction efficiency and qPCR, a non-treated muscle was spiked with the pDNA constructs before homogenization.”

Results, page 6, beginning of 2nd paragraph of Results section: Is it the levels of plasmid DNA at 24 hrs afters electroporation? If that is so, then what are the values of the positive control and the values of pDNA from 24 hrs to 84 days with respect to the positive control?

The positive control mentioned in this paragraph is not related to the treatment itself. It is only a positive control for pDNA extraction efficiency and qPCR. Therefore, it would not be relevant to quantify the results of the pDNA levels in mice to the levels of the positive control. As we wanted to stress the evolution of the pDNA levels over time, we chose to take one animal in the earliest timepoint group as a reference animal and quantified all other data compared to this reference.

In the discussion, the authors state that - "In the current study, the initial pDNA loss (within 24 hours) was similar to the pDNA loss within 48 hours in the study by Cappelletti et al. [17]." which is about a 10000 fold decrease. There is no data shown to support this claim. Was the initial 10000-fold pDNA loss at 24 hours evaluated using positive control?

Data supporting this claim is not shown in the paper as we want to focus on the evolution of the pDNA levels over time. In some preliminary experiments, we did an absolute quantification of the pDNA that we could detect 24 hours post treatment and compared this to the amount that we injected, like they do in the paper of Cappelletti et al. Most likely a major part of the injected pDNA is lost immediately after injection by leakage out of the muscle, within the first 24 hour period. However, this is not experimentally confirmed.

(I believe it would not be worthwile to evaluate levels of pDNA in negative control as there should be no DNA to amplify using qPCR).

Indeed, qPCR of negative controls gave no results. Therefore, these datapoints are not shown.

Page 8, last line of the text just before Figure 4: "major part of the nuclei" or majority of the nuclei?

Thank you for pointing out this miswording. We want to state that the majority of the nuclei does not contain any pDNA copy. The text is adapted accordingly (Line 341)

Page 9, Figure 5: It might be helpful to clarify what the symbols mean in the figure (square, diamond, circle..).

For each muscle, 6 tissue slices were investigated. These 6 slices from the same muscle and thus the same animal, are displayed with the same symbol. To clarify this, the figure legend of Figure 5 was adapted (Line 368-373):

“(A) DNA scope nuclear pDNA, data points indicate the percentage of DNA dots in nuclei in a single tissue slice, data points from tissue slices originating from the same muscle are indicated with the same symbol (3 animals per time point). (B) DNA scope DNA spot numbers, data points indicate the amount of DNA spots in a single tissue slice, data points from tissue slices originating from the same muscle are indicated with the same symbol (3 animals per time point).”

It is important to give the absolute no. of DNA spots in the nuclei. Will it be constant over time as is shown for quantified pDNA in the nucleii relative to the control at 24 h in Figure 6? The relation/percentage given now can be interpreted by the decrease pf DNA spots in the cytosol being larger than in the nuclei; but it can also be a result of increasing (in absolute terms) of DNA spots in nuclei. Which probably is not the case, but authors need to provide absolute values in order to avoid confusion and potential misinterpretation.

As only a small part of the tissue is evaluated, giving the absolute number of spots would be misleading. For example, if a tissue slice is more closely located to the injection site, a higher number of DNA spots will be present, probably also a higher absolute number in nuclei which would not necessarily mean that there is overall a higher amount of pDNA in the nuclei. Therefore, we chose to only give percentages of nuclear vs total pDNA in the regions investigated.

To cope with the limited regions evaluated with DNA scope. Nuclei were extracted and nuclear pDNA quantified for two timepoints. The qPCR data of nuclear pDNA at these timepoints (figure 6) confirm the hypothesis that the nuclear pDNA stays stable over time and that the increasing percentage of pDNA in the nuclei is due to a larger decrease of DNA in cytosol vs nuclei.

Further, it might also be useful to give the total no. of nucleii containing DNA as as percentage of total no. of nucleii for 24 h, W1 and W6. This will give some estimates of the no. of cells containing pDNA to express mAb compared to total no. of cells. (E.g. Less than x% of the cells express the mAb gene. The assumption here is that the whole muscle was exposed to the electric field.)

The DNA scope data only evaluate a very small part of the muscle, therefore making claims about the number of nuclei containing pDNA is not possible. Also, muscle cells are multinucleated, implying that the % of nuclei containing pDNA does not necessarily implies the % of cells expressing mAb. 

Discussion: Few things missing from the discussion are: -

  1. mRNA levels showed no significance decrease in 12-week follow-up, whereas plasma mAB concentrations increased up to day 14, remained constant upto day 21 and then decreased over the next 9 weeks. How do the authors explain this? Why is there a decrease in plasma conc. levels when the mRNA levels remain more or less constant.

The reason for this is still unclear and further research would be needed to clarify this apparent discrepancy.

  1. The authors claim that pDNA level in the nucleii remain stable based on observations from day 4 and day 14. Additional information can be obtained from DNA scope if authors can determine the absolute no. of pDNA spots in nucleii on day 1, w1 and w6. This will perhaps help clarify the fate of pDNA in nucleii till w6.

As indicated above, relying on the absolute number of pDNA spots in nuclei could result in misinterpretation of the actual situation. Therefore, in our opinion we cannot conclude if the pDNA levels remain stable up to week 6 from the data obtained in this study.

Can the authors comment on the fate of pDNA in nucleii. Does the amount of nuclear pDNA decrease over time (beyond 14)? How does this correspond to stable mRNA levels? And how does this correspond to decreasing plasma mAB levels beyond day 21?

The pDNA amount in nuclei after day 14 is not evaluated quantitatively for whole muscle tissue, only a fraction of the tissue is evaluated with DNA scope. Therefore, no strong claims can be made about the total nuclear pDNA content after day 14. However, the total pDNA level at W6 and W12 is slightly lower than the nuclear pDNA level at day 14, which implies that eventually over time the nuclear pDNA level will probably also decrease due to cell turnover. This would logically result in a slight decrease in mRNA levels and eventually a decrease in protein levels. In the data obtained in this study this decrease in mRNA levels is not clearly observed but the variation in mRNA levels between different animals is large. Therefore, small changes in mRNA levels over time are difficult to observe, preventing us from making strong statements in that respect.

Discussion, page 11, 2nd paragraph: authors state: “In the current study, the initial loss (within 24 hours) was similar to the pDNA loss within 48 hours in the study by Cappeletti et al” - Where is the data to support this claim? Was the initial 10000-fold pDNA loss at 24 hours evaluated using positive control?

Data supporting this claim is not shown in the paper as we want to focus on the evolution of the pDNA levels over time. In some preliminary experiments, we did an absolute quantification of the pDNA that we could detect 24 hours post treatment and compared this to the amount that we injected, like they do in the paper of Cappelletti et al. Most likely, a major part of the injected pDNA is already lost immediately after injection by leakage out of the muscle, within the first 24 hour period. However, this is not experimentally confirmed. Minor adjustment was made to the text of the discussion, to clearly state that the decrease is compared to the injected amount of pDNA. (Line 435-439):

“A previous study by Cappelletti et al. [17] showed that within 48 hours post-injection the pDNA levels decreased 10 000-fold compared to the injected dose. No pDNA quantification was performed at later time points. In the current study, the initial pDNA loss (within 24 hours) appears to be similar to the pDNA loss within 48 hours in the study by Cappelletti et al. [17].”

Reviewer 2 Report

There are many advantages to transfer the interesting gene both in vitro and in vivo by DNA plasmid. However, the lower expressing level limits its application. It is of great importance to optimize the plasmid design by establishing the methods to analyze and evaluate the factors affecting transgene expression. Therefore, this manuscript investigated both the plasma protein expression and the underlying mechanisms likely responsible for the transgene expression, such as the pDNA levels, target gene-encoding mRNA levels, and pDNA localization in cytoplasm and in nucleus, following intramuscular electroporation in mice. Generally, the construct, methods and the data of this manuscript are reasonable and convincing. I think it is qualified and hence it is suitable to be published.

Minor concerns:

1. page 3, section 2.3: “0,4 U/µL hyaluronidase”, 0.4?

2. page 4, section 2.8: “(.RT)-qPCR”, deletion of the full stop

3. page 6, Figure 1: “pDNA quantification..”, deletion of the full stop

4. page 6, Figure 1: it is difficult to differentiate per animal data points of pCAG-4D5-HC from pCAG-4D5-LC.

5. page 7, Figure 3: it is suggested to add 4D5 to the annotation of Y axis.

6. page 9, Figure 5B: the data points in figure A from 6 slices, from 9 slices in B? if so, please introduce it in the corresponding section of methods.

Author Response

Answers to the reviewer's questions are provided below in bold. If changes were made in the manuscript, the adjusted text and line numbers are provided. In the revised manuscript, ‘track-changes’ was used to facilitate the review process.

There are many advantages to transfer the interesting gene both in vitro and in vivo by DNA plasmid. However, the lower expressing level limits its application. It is of great importance to optimize the plasmid design by establishing the methods to analyze and evaluate the factors affecting transgene expression. Therefore, this manuscript investigated both the plasma protein expression and the underlying mechanisms likely responsible for the transgene expression, such as the pDNA levels, target gene-encoding mRNA levels, and pDNA localization in cytoplasm and in nucleus, following intramuscular electroporation in mice. Generally, the construct, methods and the data of this manuscript are reasonable and convincing. I think it is qualified and hence it is suitable to be published.

Minor concerns:

  1. page 3, section 2.3: “0,4 U/µL hyaluronidase”, 0.4?

Thank you for noticing this error, we have corrected it. (Line 107)

  1. page 4, section 2.8: “(.RT)-qPCR”, deletion of the full stop

Thank you for pointing out this mistake. It has been corrected. (Line 165)

  1. page 6, Figure 1: “pDNA quantification..”, deletion of the full stop

Error has been corrected. Thank you for noticing. (Line 286)

  1. page 6, Figure 1: it is difficult to differentiate per animal data points of pCAG-4D5-HC from pCAG-4D5-LC.

During manuscript preparation different visualizations of the data were tried but the data points are really close to each other, so even using different colors does not help in differentiation between HC and LC. In figure 2, mRNA levels, the different symbols are useful to see that there is variation between HC and LC, but that the levels of the HC and LC for one animal are similar. To have a similar formatting for all graphs, the same symbols were used for figure 1.

  1. page 7, Figure 3: it is suggested to add 4D5 to the annotation of Y axis.

Good suggestion. Naming of the Y axis is changed to “4D5 Plasma Concentrations (µg/mL)” (Line 323)

  1. page 9, Figure 5B: the data points in figure A from 6 slices, from 9 slices in B? if so, please introduce it in the corresponding section of methods.

Figure B has double the amount of data points per timepoint because the absolute number of spots counted for the heavy chain and for the light chain is displayed as a separate data point. Both the data in A and B is the result of the analysis of 6 slices per muscle/animal.

Reviewer 3 Report

1) Abstract. Muscle biopsies and blood samples were taken at different time points (up to 3 months). In muscle, pDNA levels decreased 90% between 24 hours and one week post treatment. In contrast, mRNA levels remained stable over time. 4D5 antibody plasma concentrations reached peak levels at week two followed by a slow decrease (50% after 12 weeks). Evaluation of pDNA localization revealed that extranuclear pDNA was cleared fast whereas the nuclear fraction remained relatively stable. Please clarify the this paragraph and support the sentences with the most important statistically significant values.

2) Abstract. This is in line with the observed mRNA and protein levels over time, and indicates that only a minor fraction of the administered pDNA is ultimately responsible for the observed systemic mAb levels. Therefore, efforts to increase the protein levels upon pDNA-based gene therapy should focus on strategies to increase both cellular entry and migration of the pDNA into the nucleus. The currently applied methodology can be used to guide the design and evaluation of novel plasmid-based vectors or alternative delivery methods, in order to achieve a robust and prolonged protein expression. Please clarify the conclusions.

3) The current study aims to characterize the in vivo expression of antibodies after pDNA administration on a broader basis than only plasma protein expression. Investigating pDNA levels, mAb mRNA levels, and pDNA localization will lead to a better understanding of the different factors driving protein expression. The findings of this study can contribute to the optimization of DNA-based mAb delivery. On top of that, the techniques described can also be used as a toolbox for evaluating future plasmid constructs or alternative DNA-based delivery methods. Please, underline the novelty of the study.

4) 2.12. Statistics At the start of experiments, mice were randomized based on body weight. DNA and protein quantification data are reported in a descriptive manner. RNA quantification and DNA scope data are analyzed using one-way ANOVA, when data is normally distributed (Shapiro-Wilk test), or Kruskal-Wallis test, when this is not the case, with Tukey’s/Dunn’s multiple comparisons test respectively. Nuclei pDNA quantification data is analyzed with unpaired t-tests. Statistical analysis and figure drawing were done using GraphPad Prism 9.3.1 (GraphPad Software, San Diego, California). Please add the statistically. Please add the statistically significant p-value and how the data were reported and why the different statistical tests were chosen.

5) 3. Results. Please, underline in the manuscript the most important statistically significant values to support the sentences.

6) 4. Discussion Antibody gene transfer is a promising alternative for conventional antibody treatment. Despite being under clinical evaluation, little is currently known about the fate of the injected pDNA after intramuscular electroporation. In the context of pDNA vaccination, pDNA biodistribution following intramuscular electroporation is well characterized both pre-clinically and clinically [14–16]. Please, summarise here the most important resultts of the study.

7) In addition to an elaborate investigation of the fate of pDNA after antibody gene transfer, this study measured the mRNA and protein levels in the same subjects. mRNA levels show huge variability, but no significant difference is observed between different time points. This contributes to the hypothesis that the amount of pDNA in the nuclei stays quite stable over time. The pharmacokinetic profile of the protein expression shows an initial increase in antibody titers up to week 3, followed by a steady decline in plasma protein levels. In conclusion, this study demonstrates that durable expression is dependent on the nuclear uptake of the pDNA and that efforts to increase the protein levels upon pDNA based gene therapy should also focus at strategies to increase both cellular entry and migration of the pDNA in to the nucleus. The generated toolbox can be used to guide the design and evaluation of future pDNA constructs or delivery methods, to achieve a robust and prolonged protein expression. Please, underline the limits and the possible clinical implications of the study.

Author Response

Answers to the reviewer's questions are provided below in bold. If changes were made in the manuscript, the adjusted text and line numbers are provided. In the revised manuscript, ‘track-changes’ was used to facilitate the review process.

1) Abstract. Muscle biopsies and blood samples were taken at different time points (up to 3 months). In muscle, pDNA levels decreased 90% between 24 hours and one week post treatment. In contrast, mRNA levels remained stable over time. 4D5 antibody plasma concentrations reached peak levels at week two followed by a slow decrease (50% after 12 weeks). Evaluation of pDNA localization revealed that extranuclear pDNA was cleared fast whereas the nuclear fraction remained relatively stable. Please clarify the this paragraph and support the sentences with the most important statistically significant values.

As requested statistically significant values were added for the pDNA and protein levels. (Line 20-22)

2) Abstract. This is in line with the observed mRNA and protein levels over time, and indicates that only a minor fraction of the administered pDNA is ultimately responsible for the observed systemic mAb levels. Therefore, efforts to increase the protein levels upon pDNA-based gene therapy should focus on strategies to increase both cellular entry and migration of the pDNA into the nucleus. The currently applied methodology can be used to guide the design and evaluation of novel plasmid-based vectors or alternative delivery methods, in order to achieve a robust and prolonged protein expression. Please clarify the conclusions.

The main conclusion of this study is that durable expression of mAb is dependent on the nuclear uptake of the pDNA. It is a good suggestion to clearly mention this in the abstract. (Line 26-32):

“In conclusion, this study demonstrates that durable expression is dependent on the nuclear uptake of the pDNA. Therefore, efforts to increase the protein levels upon pDNA-based gene therapy should focus on strategies to increase both cellular entry and migration of the pDNA into the nucleus. The currently applied methodology can be used to guide the design and evaluation of novel plasmid-based vectors or alternative delivery methods, in order to achieve a robust and prolonged protein expression.”

3) The current study aims to characterize the in vivo expression of antibodies after pDNA administration on a broader basis than only plasma protein expression. Investigating pDNA levels, mAb mRNA levels, and pDNA localization will lead to a better understanding of the different factors driving protein expression. The findings of this study can contribute to the optimization of DNA-based mAb delivery. On top of that, the techniques described can also be used as a toolbox for evaluating future plasmid constructs or alternative DNA-based delivery methods. Please, underline the novelty of the study.

We are the first to evaluate antibody gene transfer based on pDNA levels/localization and mRNA levels over an extensive time period post treatment. These data give unique insights in the factors driving protein expression. Understanding these factors will help to guide future optimization of antibody gene transfer. To underline the novelty in this study, the text was slightly adapted. (Line 80-87):

“The current study aims to characterize the in vivo expression of antibodies after pDNA administration on a broader basis than only plasma protein expression. To the best of our knowledge, it is the first study to investigate pDNA levels, mAb mRNA levels, and pDNA localization over an extensive time period post treatment. This will lead to a better understanding of the different factors driving protein expression. The findings of this study can contribute to the optimization of DNA-based mAb delivery. On top of that, the techniques described can also be used as a toolbox for evaluating future plasmid constructs or alternative DNA-based delivery methods.”

4) 2.12. Statistics At the start of experiments, mice were randomized based on body weight. DNA and protein quantification data are reported in a descriptive manner. RNA quantification and DNA scope data are analyzed using one-way ANOVA, when data is normally distributed (Shapiro-Wilk test), or Kruskal-Wallis test, when this is not the case, with Tukey’s/Dunn’s multiple comparisons test respectively. Nuclei pDNA quantification data is analyzed with unpaired t-tests. Statistical analysis and figure drawing were done using GraphPad Prism 9.3.1 (GraphPad Software, San Diego, California). Please add the statistically. Please add the statistically significant p-value and how the data were reported and why the different statistical tests were chosen.

To clarify the statistical part of this study, section ‘2.12. Statistics’ in Materials & Methods was expanded. Statistically significant p-value, a general description of the data reporting and motivation for the choice of statistical tests was added. (Line 253-264)

“At the start of experiments, mice were randomized based on body weight. DNA/RNA quantification and DNA scope data are available from multiple timepoints. To compare these timepoints, data are analyzed using one-way ANOVA, when data is normally distributed (Shapiro-Wilk test), or Kruskal-Wallis test, when this is not the case, with Tukey’s/Dunn’s multiple comparisons test. The decrease of plasma 4D5 levels from week 3 onwards was analyzed using a linear mixed model in R. Nuclei pDNA quantification data is analyzed with unpaired t-tests, to investigate a possible difference between the two timepoints. Data with p-value below 0.05 are considered as statistically significantly different. Statistical analysis and figure drawing were done using GraphPad Prism 9.3.1 (GraphPad Software, San Diego, California). All graphs display individual data points per animal or tissue slice and median values or average ± SD (specified in figure legends). ”

5) 3. Results. Please, underline in the manuscript the most important statistically significant values to support the sentences.

Statistically significant values were added for the pDNA quantification and protein levels. (Line 280-284/Line 313-316)

“pCAG-4D5-HC and pCAG-4D5-LC data were normally distributed and subject to one-way ANOVA combined with Dunn’s test for multiple comparisons. A significant difference in mean (i.e. a decrease of pDNA over time) was observed between 24 hours and all other timepoints (day 4: p = 0.029, other timepoints: p < 0.0001).”

“Using a linear mixed model we could demonstrate that 4D5 plasma levels changed significantly between day 21 (peak level) and day 84 (t(16.3) = -5.13, p = 9.43e-05), with significant changes when comparing to day 21 for both day 42 (t(15) = -2.43, p = 0.0273) and day 84 (t(15) = -5.585, p = 4.14e-05).”

6) 4. Discussion Antibody gene transfer is a promising alternative for conventional antibody treatment. Despite being under clinical evaluation, little is currently known about the fate of the injected pDNA after intramuscular electroporation. In the context of pDNA vaccination, pDNA biodistribution following intramuscular electroporation is well characterized both pre-clinically and clinically [14–16]. Please, summarise here the most important results of the study.

We appreciate the suggestion of the reviewer to summarize the results of our study at the beginning of the discussion. Therefore, the most important findings of the study are now added to the second paragraph of the discussion (Line 428-434):

“This study characterizes antibody gene transfer from a fundamental point of view. DNA, the starting point of the central dogma of molecular biology, is injected as part of the treatment. In muscle, pDNA levels decreased 90 % between 24 hours and one week post treatment. In contrast, mRNA levels remained stable over time. 4D5 antibody plasma concentrations reached peak levels at week two followed by a slow decrease (50 % after 12 weeks). Evaluation of pDNA localization revealed that extranuclear pDNA was cleared fast whereas the nuclear fraction remained relatively stable. This is in line with the observed mRNA and protein levels over time, and indicates that only a minor fraction of the administered pDNA is ultimately responsible for the observed systemic mAb levels.”

7) In addition to an elaborate investigation of the fate of pDNA after antibody gene transfer, this study measured the mRNA and protein levels in the same subjects. mRNA levels show huge variability, but no significant difference is observed between different time points. This contributes to the hypothesis that the amount of pDNA in the nuclei stays quite stable over time. The pharmacokinetic profile of the protein expression shows an initial increase in antibody titers up to week 3, followed by a steady decline in plasma protein levels. In conclusion, this study demonstrates that durable expression is dependent on the nuclear uptake of the pDNA and that efforts to increase the protein levels upon pDNA based gene therapy should also focus at strategies to increase both cellular entry and migration of the pDNA in to the nucleus. The generated toolbox can be used to guide the design and evaluation of future pDNA constructs or delivery methods, to achieve a robust and prolonged protein expression. Please, underline the limits and the possible clinical implications of the study.

The main limitation of this study is the difficulty to make correlations between DNA/RNA and protein levels. This is due to several factors, (1) mice are treated in both tibialis anterior muscles: only one is used for pDNA and mRNA quantification, while both contribute to protein expression; (2) isolating muscles for pDNA and mRNA extraction is a terminal procedure, no further plasma protein level follow-up is possible. Nevertheless, it is likely that pDNA/mRNA levels correlate with plasma protein levels at a later timepoint given the plasma half-life of antibodies. These limitations of the study are added to the discussion. (Line 533-543)

“Unfortunately, the data of this study do not allow to directly correlate the protein expression levels to pDNA and mRNA levels because of several factors, (1) mice are treated in both tibialis anterior muscles: only one is used for pDNA and mRNA quantification, while both contribute to protein expression; (2) isolating muscles for pDNA and mRNA extraction is a terminal procedure, no further plasma protein level follow-up in the same animals is possible. Nevertheless, it is likely that pDNA/mRNA levels correlate with plasma protein levels at a later timepoint given the plasma half-life of antibodies. The percentage of nuclear pDNA and the number of pDNA dots of the DNA scope assay do not show a correlation with protein expression either. However, as pointed out before, this data was extracted from one small part of one of two treated muscles, so this experiment cannot exclude the possible correlation either.”

The clinical implications of this study are limited as it is fundamental research. The findings of this study will help to improve antibody gene transfer, which could in the longer run result in clinical therapies. This fundamental study also developed a toolbox which can be used to thoroughly evaluate future plasmid constructs or delivery methods. As stated in the conclusions of this paper (Line 547-549).
